# Protein-Coding Genes in Euarchontoglires with Pseudogene Homologs in Humans

**DOI:** 10.3390/life10090192

**Published:** 2020-09-10

**Authors:** Lev I. Rubanov, Oleg A. Zverkov, Gregory A. Shilovsky, Alexandr V. Seliverstov, Vassily A. Lyubetsky

**Affiliations:** 1Institute for Information Transmission Problems of the Russian Academy of Sciences (Kharkevich Institute), Moscow 127051, Russia; rubanov@iitp.ru (L.I.R.); zverkov@iitp.ru (O.A.Z.); gregory_sh@list.ru (G.A.S.); slvstv@iitp.ru (A.V.S.); 2Belozersky Institute of Physico-Chemical Biology, Lomonosov Moscow State University, Moscow 119234, Russia; 3Faculty of Biology, Lomonosov Moscow State University, Moscow 119192, Russia

**Keywords:** recently pseudogenized genes, human pseudogenes, efficient software, independent pseudogenization in hominoids, Euarchontoglires group

## Abstract

An original bioinformatics technique is developed to identify the protein-coding genes in rodents, lagomorphs and nonhuman primates that are pseudogenized in humans. The method is based on per-gene verification of local synteny, similarity of exon-intronic structures and orthology in a set of genomes. It is applicable to any genome set, even with the number of genomes exceeding 100, and efficiently implemented using fast computer software. Only 50 evolutionary recent human pseudogenes were predicted. Their functional homologs in model species are often associated with the immune system or digestion and mainly express in the testes. According to current evidence, knockout of most of these genes leads to an abnormal phenotype. Some genes were pseudogenized or lost independently in human and nonhuman hominoids.

## 1. Introduction

Pseudogenes were for long relegated to the “junk” portion of DNA, but nowadays they attract an increasing attention for bearing important biological functions [1,2], e.g., in gene expression regulation and development of human diseases. Association of pseudogenes with human disorders has been approached in many works. Among numerous examples are interfaces in cancer [3,4,5,6], type 2 diabetes [7], pulmonary fibrosis, adrenal hyperplasia, chronic pancreatitis, AIDS and others [1]. With many pseudogenes already known from human and model species [8], research continues towards an effective computer technique for large-scale prediction of pseudogenes with common protein-coding homologs in a wide set of species [9].

The challenge is to detect pseudogenes in a species of interest (in our case, humans) or a set of species based on ancestry patterns of protein-coding homologs in a query species set (here, the Euarchontoglires group).

## 2. Materials and Methods

We searched for the protein-coding genes in 3 reference species, namely, the mouse (*Mus musculus*), rat (*Rattus norvegicus*) and rabbit (*Oryctolagus cuniculus*), which are present in at least 4 of 5 nonhuman hominoids, at least 2 of 12 Old World monkeys and pseudogenized in humans. Figure 1 shows the phylogenetic tree indicating the lineages that were queried to discover such genes.

The solution was realized by verifying all the protein-coding genes *X* in the reference species against genes *Y*, including pseudogenes, in a given species set (here, 18 hominoids and cercopithecoids, including humans). The verification includes concurrent verifications of synteny in the neighborhoods of *X* and *Y* and gene orthology; pseudogene *Y* is verified for the same local synteny, homology and conservation of the ancestral exon-intronic structure. Such *X* and *Y* genes are referred to as consistent. The method allows for a reasonable parameter choice. For example, the neighborhood size is defined by a typical length of the topologically associated domain [10]. For nonhuman primates, the predicted consistency was verified in the neighborhood of 2 Mb under the constraining presence of at least two pairs of protein-coding one-to-one orthologs different from each other and the candidate gene; the neighborhood was 5 Mb and at least 4 (occasionally 3) such pairs were required for humans. Such orthologs are defined as witnesses to the candidate genes. The main stages of the method are efficiently implemented using fast computer software [11] that allows joint processing of over 100 complete genomes in a reasonable time. This program is an essential enhancement of our previous software once used in [12,13]. Genomic data were retrieved from Ensembl v99 [14], and data on the organ-specific expression of genes from Expression Atlas [15].

The results reported in this short communication were obtained with the following procedure (Figure 2). Initially, the program lossgainRSL was used to infer list *A* of the protein-coding genes in each reference species, which exist in the overwhelming majority of the nonhuman hominoids and many cercopithecoids but are lost or pseudogenized in human. The presence or absence of a gene is determined from orthology and local synteny: as stated above, at least two witnesses are required in the gene’s neighborhood. The list is then pruned: for each gene *X*, its neighborhood of at most 1 Mb is verified on the genomic alignment with the human genome. The neighborhood is verified for the presence of at least four witness genes having one-to-one orthologs in the counterpart human region. The gene *X-*aligned region in the human is then considered. Three cases are examined: (1) exonic gene *X* sequences hitting the noncoding regions in humans; (2) human regions are protein-coding but non-orthologous to gene *X*; and (3) human regions contain a pseudogene. Pseudogene types (processed, unprocessed, unitary, etc.) may be analyzed or chosen on a setting basis. In this work, all types are treated equally. In Case (1), gene *X* is assumed as lost in humans and removed from list *A*. In Case (2), the gene’s organ-specific expression and exon-intronic structural patterns are compared between the reference species and human. If similarity is detected between any of the patterns, non-orthology is attributed to a low exonic similarity; the gene is not assumed as lost and also omitted from list *A*. In Case (3), a human pseudogene rather strongly aligns only to some exons of gene *X*, thus suggesting its pseudogenization; such genes are kept in list *A* and build a list of the reference species genes that are pseudogenized in human. Naturally implied parameters of this selection were estimated empirically.

Pseudogene lists generated with this method are united across the three reference species. The lists can instead intersect or vote by reference species. With many reference species, voting is superior. Prior inference of list *A* is a key step for reducing the dimension of the reference species gene entries from tens of thousands to several hundreds, which speeds up the downstream analyses by two orders of magnitude and enables varying the computationally heavy parameters.

## 3. Results

### 3.1. Consistent Pseudogene Gene Pairs Identified

Our technique isolated only 50 pseudogenes in humans: 42 consistent with mouse genes, 42 of slightly various content with the rat and 38 with the rabbit (Table 1). An expanded version of the table is provided in Appendix A, where the human pseudogenes are described in Columns A–H, their coding consistencies in the mouse in Columns I–R, in the rat in Columns S–AB and in the rabbit in Columns AC–AK. Columns Q and R contain the species numbers of the nonhuman hominoids and Old World monkeys, respectively—found to have a mouse ortholog in the current row under the imposed conditions (including number of witnesses). The numbers for the rat and rabbit are provided in Columns AA–AB and AJ–AK, analogously.

Reference species were found to possess 43 (mouse), 42 (rat) and 37 (rabbit) protein-coding genes pseudogenized in humans and some other hominoids. Among those, 10 genes in the mouse (Z*p3r, Prss40, Prss46, Tmem30c, Dpy19l2, Adam5, Gm4787, Hils1, 4931406B18Rik,* and *Cst13*) and 13 in the rat (*Clca4l, Zp3r, Prss40, Prss46, Tmem30c, Taar4, Dpy19l2, Adam5, Adam4, Hils1, Cyp2g1, LOC690483*, and *Cst13*) are expressed almost exclusively in the testes. Another 9 murine and 15 rat genes are highly expressed in the testes, but also active in other organs (*H2bu2, Prorsd1, Ahsa2, Htr5b, Pcdhb14, Ccdc162, Tdh, Slc22a20, Zfp35, Clca5*, *Cym, Hist3h2ba, Prorsd1, Ahsa2, Tmed11, Fbxl21, Pcdhb14, Tdh, Olr836, Slc22a20, Tmem198b, Mettl21cl1, Zfp35* and *RGD1560171,* respectively).

Among the human pseudogenes in Appendix A, only one is annotated as processed, whilst the others represent transcribed unitary pseudogenes (21), transcribed unprocessed pseudogenes (17), unprocessed pseudogenes (8) and unitary pseudogenes (3). Accordingly, a total of 50 pseudogenes are predicted, with one of them listed three times in Appendix A.

Many of the identified genes are associated with the immune system, such is the *Zfp35* gene, whose knockout in mice results in an abnormal T-helper 2 cell differentiation, increased airway responsiveness, increased circulating interleukin-13 level, increased circulating interleukin-4 level, increased circulating interleukin-5 level and increased eosinophil cell number [16]. This gene is expressed mainly in the testes, but also maintains high levels in the thymus, spleen and brain. In humans it is represented by the unitary transcribed pseudogene *ZNF271P*. Some detected protein-coding genes are involved in digestion, e.g., murine chymosin-encoding *Cym*.

Available evidence shows that knockout of many identified human pseudogene homologs in model species leads to severe disorders. For example, mice with a human-like *Cmah* deficiency have hyperactive macrophages, T cells, B cells, etc. [17]. Knockout of the heat shock protein ATPase 2 activator gene *Ahsa2* leads to abnormal cornea morphology and decreased total retina thickness [18], whilst urate oxidase-encoding *Uox* knockout causes abnormal kidney morphology and uremia [19]. Knocking out gene *Tmem198b* of the pituitary gland transmembrane protein 198b provokes reduced sensorimotor gating [20]. In contrast, two mouse genes, *4931406B18Rik* and *Htr5b*, exhibit no abnormal phenotype in ablation [21].

Of interest is the identified F-box and leucine-rich repeating protein 21-encoding gene *Fbxl21*. In mice and rats it is highly expressed within the suprachiasmatic nuclei, the site of the master clock, where it displays marked circadian oscillations apparently driven by members of the PAR-bZIP family [22]. Its knockout is associated with a shortened circadian behavioral period through ubiquitination and stabilization of cryptochromes [23], limb grasping and decreased grip strength [18]. In humans, it is consistent with the pseudogene *FBXL21P* transcribed in the brain, kidneys and prostate, and the pseudogene ENSGGOG00000014124 in the gorilla. In the gibbon, chimp, bonobo and orangutan its coding homolog is preserved in the same syntenic context. Pseudogenization of this circadian rhythm regulator may presumably be associated with increased longevity, as well as emerging neoteny in humans [24]. Gui et al. [25] studied the impact of SNPs in *FBXL21* on the success of a kidney transplantation.

The identified *Ctf2, Cyp2g1, Fmo6, Olfr155, Olfr159, Olfr433* and *Taar4* genes are highly expressed in the vomeronasal organ or olfactory epithelium, which come in concordance with its reduction and degraded olfaction in humans [24].

Homologs of the rat gene *RGD1560171* survived in all five of the non-human hominoids studied and many short-living mammals except mice (pseudogene *Gm715*). This protein-coding gene is pseudogenized or lost in species with a relatively high longevity, including humans (*AL589987.1*), naked mole rats and elephants. The gene *Ofcc1* in the mouse, *AABR07027339.1* in the rat and ENSOCUG00000016635 in the rabbit are consistent with the human pseudogene *OFCC1* and belong to same family with a gene potentially causal for orofacial cleft in humans [26]. However, ablation of this gene had no effects in head development in mice.

Each human pseudogene is consistent with exactly one gene in mouse, except for *CLCA3P*, consistent with three syntenically linked paralogs in mice and two in rats. Each human pseudogene is consistent with exactly one protein-coding gene in the rabbit.

The human pseudogene homologs of 16 murine protein-coding genes are presumably functional in the five non-human hominoids, whilst other murine genes survived in only four species. The same ratio is observed with other reference species. Human pseudogenes presumably retain functionality in many Old World monkeys and other placental mammals (Appendix A).

### 3.2. Consistent Genes of the Reference Species and Their Counterparts in Other Species

Predictions in nonhuman primates and other placental mammals against the mouse, rat and rabbit are given in Appendix A, respectively. The species column contains gene IDs if a gene is present in the species. Mouse genes were found on average in 4.4 nonhuman hominoids (17 genes per 5 species) and 9.6 Old World monkeys; rat genes—in 4.4 nonhuman hominoids (17 genes per 5 species) and 9.4 Old World monkeys; and rabbit genes—in 4.4 nonhuman hominoids (15 genes per 5 species) and 9.5 Old World monkeys.

The mouse was found to have 43 consistencies with 42 human pseudogenes. Consistency is “one-to-one”, with the exception of the three mouse genes consistent with the same pseudogene *CLCA3P* and the mouse gene *Cyp2g1*, consistent with two neighboring human pseudogenes, *CYP2G1P* and *CYP2G2P*. Consistency is supported by local alignment of corresponding genes accounting for the exon-intronic structure and genomic alignment with at least 4 pairs of orthologous witnesses in a neighborhood of a specified size (3 pairs in the sole case), with the pair numbers normally being much greater. Appendix A describes the witnesses; pseudogenes are shadowed green and the syntenic blocks separated by horizontal lines. The nearest protein-coding genes flanking the candidate genes, where possible, were chosen as witnesses within the specified neighborhoods.

The rat was predicted to have 42 genes consistent with 42 human pseudogenes. Consistency is “one-to-one”, with the exception of two genes in the rat consistent with the same human pseudogene *CLCA3P* and the rat gene *Cyp2g1*, consistent with two syntenic human pseudogenes, *CYP2G1P* and *CYP2G2P*. The exceptions thus mimic those in the mouse. The consistency is supported by the local alignment of the corresponding genes, accounting for the exon-intronic structure and genomic alignment with at least 4 pairs of orthologous witnesses in a neighborhood of a specified size (3 pairs in two cases), with the pair numbers normally being much greater. Appendix A describes the witnesses; pseudogenes are shadowed green and the syntenic blocks separated by horizontal lines.

Rabbit was found to have 37 genes consistent with 38 human pseudogenes. Consistency is “one-to-one”, with the exception of the rabbit gene ENSOCUG00000005745, consistent with same syntenic human pseudogenes, *CYP2G1P* and *CYP2G2P*. Consistency is supported by the local alignment of the corresponding genes accounting for the exon-intronic structure and genomic alignment with at least 4 pairs of orthologous witnesses in a neighborhood of a specified size (3 pairs in the sole case), with the pair numbers normally being much greater. Appendix A describes the witnesses; pseudogenes are shadowed green and the syntenic blocks separated by horizontal lines.

In most cases, the detected pseudogenes are confined strictly within a syntenic region, with the following 10 outliers found at the boundary: *A2MP1, AL589987.1, CST13P, CYP2G1P, CYP2G2P*, *H2BU2P, KLRA1P, METTL21EP, OFCC1* and *TMED11P*; see Appendix A.

### 3.3. Human Pseudogenes Independently Pseudogenized or Lost in Exactly One Nonhuman Hominoid

In contrast to humans, a few pseudogenes are known from nonhuman primates: 71% (human), 2% (gorilla), 3% (bonobo), 2% (chimpanzee), 5% (orangutan) and 62% (mouse) of the total protein-coding genes. The lack of some inferred genes in exactly one nonhuman hominoid may indicate their evolution into pseudogenes that escaped detection.

It follows from Appendix A that the pseudogenes found in humans presumably all retain functionality in the common chimpanzee and bonobo—all except *CCDC92B, H1-9* and *SKINT1L*; in the gibbon—all except *ADAM20P1, AL160191.3, CST13P* and *TDH*; in gorilla—all except *A2MP1, ADAM5, CYP2G1P, CYP2G2P, GUCY1B2, FBXL21P* and *LINC00643*; and in the orangutan—all except for 17 genes. These 31 cases of human pseudogenes that lost function in other hominoids are provided in Appendix A (shadowed green in the first column) along with the relevant witnesses. Such gene groups are separated by horizontal lines. Pseudogenes are consistent with the reference species genes specified in Appendix A that lack in exactly one nonhuman hominoid (bonobo, gorilla, gibbon or orangutan). Each group, starting from Column I, contains consistent nonhuman hominoid genes. For every pseudogene (except two), the genomic alignment contains a region without a consistent coding gene (marked “no gene”, except for three cases in gorilla and three in orangutan). In two cases in the gorilla, the pseudogenes are consistent with known pseudogenes, and in one case—to a coding gene with no orthology in the reference species, thus introducing conflict in the consistency with the reference and human. In the orangutan, one instance lacks genomic alignment (shadowed blue), while in the other two cases a pseudogene is consistent with three coding genes and a coding gene with a pseudogene (shadowed green), respectively. In the rest of the cases, 3 in bonobo, 4 in gorilla and gibbon, 14 in orangutan, a pseudogene-consistent coding gene is not inferred in the nonhuman hominoids, although the sequence comparison sometimes suggests its possible pseudogenization. The description fields contain the gene attributes.

## 4. Discussion

More than a half (28 of 52) of the consistencies in Appendix A span the mouse, rat and rabbit, with the corresponding protein-coding genes being one-to-one orthologs with at least two orthologous witness pairs (normally more) within the 2 Mb neighborhood. The two exceptions are the already discussed pseudogene *CLCA3P*, consistent with three mouse genes, *Clca3a1*, *Clca3a2* and *Clca3b*; two rat genes, *Clca5* and *Clca4l*; and one rabbit gene, ENSOCUG00000003548. Here, the genes *Clca4l* and *Clca3b* are one-to-one orthologs; *Clca5*, *Clca3a1* and *Clca3a2*—one-to-many orthologs; and the rabbit gene is a one-to-many ortholog to the above listed three mouse genes and two rat genes. The second exception, killer cell lectin-like receptor A1 pseudogene *KLRA1P*, is consistent with the murine *Klra2*. However, its ortholog *Klra2* in the rat has no reliable genomic alignment with the *KLRA1P* neighborhood, which might be attributed to the available rat genome assembly. This pseudogene is therefore consistent with the immunoreceptor gene *Ly49i7* in the rat, non-orthologous to the murine *Klra2*. Pseudogene *KLRA1P* is also consistent with the gene ENSOCUG00000007301 in the rabbit, orthologous to 11 murine genes and 23 rat genes, including the aforementioned *Klra2* and *Ly49i7*.

Consider cases when a human pseudogene is consistent with the coding homologs in a subset of reference species; such pseudogenes are marked yellow in Appendix A. The following 4 pseudogenes have a single consistency in the rabbit, with no orthology in the mouse and rat: *ABCC13, CRYGFP, KRT43P* and *LINC00643*. Conversely, the following 8 pseudogenes have consistencies in the mouse and rat, with no orthology in the rabbit: *AC063977.5, ALOX12P2, FBXL21P, GUCY1B2, OR13C6P, OR2S1P, PCDHB17P* and *SKINT1L*. Pseudogene *OR2S1P* is consistent with the non-orthologous murine *Olfr155* and rat *Olr840* genes, both encoding olfactory receptor proteins, however. The rabbit has neither an *OR2S1P* consistency nor orthologs of *Olfr155* and *Olr840* in the rat and mouse, respectively.

The same is true with pseudogene *H2BU2P*: its consistencies in the mouse and rat are one-to-many orthologs of the gene ENSOCUG00000033562 in the rabbit. However, the rabbit gene is not consistent with the pseudogene *H2BU2P* but is a one-to-one ortholog of the active *H2BU1* gene in humans. A special case is pseudogene *TMED11P*, consistent with *Tmed11* in the mouse and rat. These are orthologs of ENSOCUG00000021044 in the rabbit that aligns to *TMED11P* but within the short, less than 70 Kb scaffold GL019412 in the available assembly, which allows no reliable consistency prediction. Recall that murine *Fbxl21* is expressed in many parts of the brain (listed in lowering order): pituitary gland, medulla oblongata, arcuate nucleus of hypothalamus, dorsal raphe nucleus, preoptic area, corpora quadrigemina, corpus striatum, diencephalon, cerebellum, spinal cord and hippocampal formation; its highest expression in the rat is also observed in the brain.

Let us describe cases where a pseudogene has no coding consistencies in the mouse or rat. Pseudogenes *PRSS40B* and *PRSS40A* neighbor in human Chromosome 2, the first being consistent with *Prss40* in the mouse and the second in the rat, both orthologs encoding serine protease 40 but hitting different pseudogenes in the genomic alignment. Pseudogene *PRSS40B* is consistent with the gene ENSOCUG00000006394 in the rabbit. A similar case is observed with pseudogenes *AL160191.3* and *ADAM20P1* co-located in human Chromosome 14 at about a 200 Kb distance, the first being consistent with the rat *Adam4* and the second to the murine *Gm4787*. These genes are one-to-many orthologs verified with the same human witnesses and both expressed mainly in the testes in these rodents. Pseudogene *ADAM20P1* in the rabbit is consistent with the gene ENSOCUG00000008563.

Pseudogene CCDC162P is consistent with the murine *Ccdc162* and rabbit ENSOCUG00000006035, with no orthologs in the rat. Alignment of this pseudogene’s neighborhood to the rat genome does not support consistency to any coding gene in the rat, although the flanking genes are orthologous (relative to the pseudogene spot in humans and a putative gene spot in rats). Variance between the available mouse and rat genomes can hence be illustrated. Pseudogene *AL589987.1* is analogously consistent with *RGD1560171* in the rat and ENSOCUG00000012140 in the rabbit, with no murine orthology. The same is true with pseudogene *OFCC1*, consistent with *AABR07027339.1* in the rat and ENSOCUG00000016635 in the rabbit, with no murine orthology. Pseudogene *OR10AA1P* is consistent with the murine *Olfr433* and ENSOCUG00000036381 in the rabbit, with no orthologs in the rat.

Specially consider the following two rows in Appendix A. Pseudogene *DPY19L2P1* finds consistency with *Dpy19l2* in the rat within one genome alignment locus verified with witnesses from Appendix A. In another alignment spot, however, this rat gene has an orthologous hit to coding gene *DPY19L2* in human Chromosome 12. Unlike with the pseudogene in Chromosome 7, this ortholog is not verified with any witness in Chromosome 12. Therefore, Appendix A shows consistency to a pseudogene and not a protein-coding gene. The same is true for *Dpy19l2* in the mouse and *DPY19L2* in the rabbit.

Pseudogene *A2MP1* is located at the boundary of a syntenic locus (Appendix A), with witnesses found within a 200 Kb neighborhood in the mouse and rat, and within 6.5 Mb in humans, thus exceeding the value of 5 Mb suitable in all other cases. Worth noticing also is that the genomic alignment of the rat alpha-2-macroglobulin-like 1 gene *A2ml1* aligns the non-orthologous ovostatin 2-encoding human gene *AC024940.7* in the alternative region CHR_HSCHR12_4_CTG2 of Chromosome 12, but not pseudogene *A2MP1* co-located with *AC024940.7*. Genome alignment was likely affected in this case by close presence of the rat pregnancy-zone protein-encoding gene *Pzp* that is also highly affine to this pseudogene. A similar pattern is observed in the mouse and rabbit. Appendix A shows the human pseudogene *A2MP1′*s consistencies with the genes in all three reference species.

## 5. Conclusions

A method is proposed for the prediction of the protein-coding genes in a large set of species that are pseudogenized or lost in a species of interest. A fast computer implementation is freely available. The method allows processing of over 100 complete genomes in a reasonable time. It was applied to predict 50 pseudogenes in humans with functional homologs in rodents, lagomorphs and primates, associated with the immune system, digestion or expressed predominantly in the testes. These genes may represent important targets in studies of human longevity and neoteny, while their co-regulation network may be involved in the development of airway responsiveness and other autoimmune disorders.

## Figures and Tables

**Figure 1 life-10-00192-f001:**
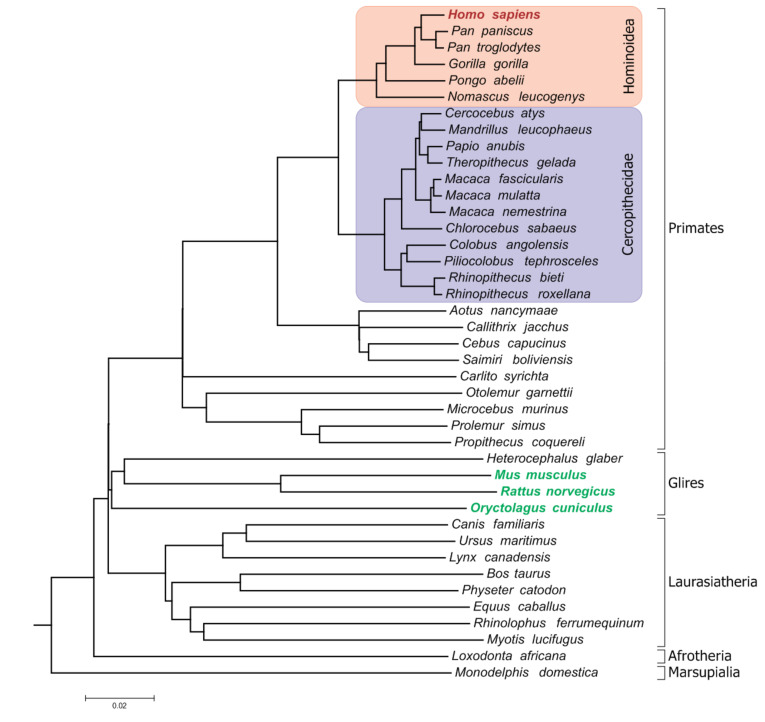
The phylogenetic tree of the species involved in the study. The reference species are shown in green. The lineages that were queried to discover the gene-loss/pseudogenization events are highlighted.

**Figure 2 life-10-00192-f002:**
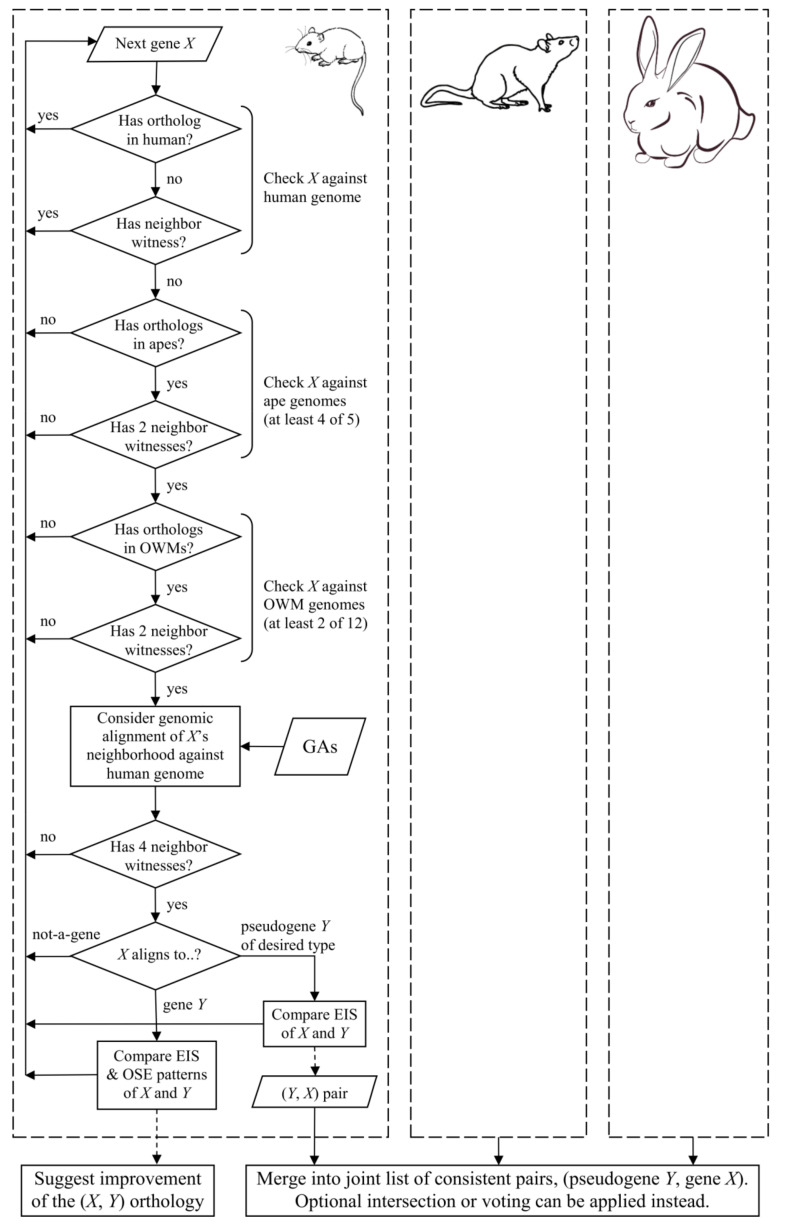
Schematic of the proposed method as applied to species and parameters involved in this study. OWM, Old World monkeys; GA, genomic alignment; EIS, exon-intronic structure; OSE, organ-specific expression. Dashed arrows signify side effects.

**Table 1 life-10-00192-t001:** Human pseudogenes and their consistencies in the mouse, rat and rabbit.

Human Pseudogene	Consistent Mouse Gene(s)	Consistent Rat Gene(s)	Consistent Rabbit Gene
*A2MP1*	*A2ml1*	*A2ml1*	ENSOCUG00000000313
*ABCC13*			ENSOCUG00000012802
*AC063977.5*	*4931406B18Rik*	*LOC690483*	
*ADAM20P1*	*Gm4787*		ENSOCUG00000008563
*ADAM5*	*Adam5*	*Adam5*	ENSOCUG00000005475
*AHSA2P*	*Ahsa2*	*Ahsa2*	ENSOCUG00000017817
*AL160191.3*		*Adam4*	
*AL589987.1*		*RGD1560171*	ENSOCUG00000012140
*ALOX12P2*	*Alox12e*	*Alox12e*	
*C4BPAP1*	*Zp3r*	*Zp3r*	ENSOCUG00000023993
*CCDC162P*	*Ccdc162*		ENSOCUG00000006035
*CCDC92B*	*Ccdc92b*	*Ccdc92b*	ENSOCUG00000027623
*CLCA3P*	*Clca3a1, Clca3a2, Clca3b*	*Clca5, Clca4l*	ENSOCUG00000003548
*CMAHP*	*Cmah*	*Cmahp*	ENSOCUG00000004430
*CRYGFP*			ENSOCUG00000010967
*CST13P*	*Cst13*	*Cst13*	ENSOCUG00000026263
*CTF2P*	*Ctf2*	*Ctf2*	ENSOCUG00000027802
*CYMP*	*Cym*	*Cym*	ENSOCUG00000026023
*CYP2G1P*	*Cyp2g1*	*Cyp2g1*	ENSOCUG00000005745
*CYP2G2P*	*Cyp2g1*	*Cyp2g1*	ENSOCUG00000005745
*DPY19L2P1*	*Dpy19l2*	*Dpy19l2*	ENSOCUG00000009890
*FBXL21P*	*Fbxl21*	*Fbxl21*	
*FER1L4*	*Fer1l4*	*Fer1l4*	ENSOCUG00000012872
*FMO6P*	*Fmo6*	*Fmo6*	ENSOCUG00000005169
*GUCY1B2*	*Gucy1b2*	*Gucy1b2*	
*H1-9*	*Hils1*	*Hils1*	ENSOCUG00000012304
*H2BU2P*	*H2bu2*	*Hist3h2ba*	
*HTR5BP*	*Htr5b*	*Htr5b*	ENSOCUG00000016884
*KLRA1P*	*Klra2*	*Ly49i7*	ENSOCUG00000007301
*KRT43P*			ENSOCUG00000011173
*LINC00643*			ENSOCUG00000026292
*METTL21EP*	*Mettl21e*	*Mettl21cl1*	ENSOCUG00000012425
*OFCC1*		*AABR07027339.1*	ENSOCUG00000016635
*OR10AA1P*	*Olfr433*		ENSOCUG00000036381
*OR13C6P*	*Olfr159*	*Olr836*	
*OR2S1P*	*Olfr155*	*Olr840*	
*PCDHB17P*	*Pcdhb14*	*Pcdhb14*	
*PRORSD1P*	*Prorsd1*	*Prorsd1*	ENSOCUG00000011673
*PRSS40A*		*Prss40*	
*PRSS40B*	*Prss40*		ENSOCUG00000006394
*PRSS46P*	*Prss46*	*Prss46*	ENSOCUG00000039130
*SKINT1L*	*Skint1*	*Skint1*	
*SLC22A20P*	*Slc22a20*	*Slc22a20*	ENSOCUG00000003132
*TAAR4P*	*Taar4*	*Taar4*	ENSOCUG00000024372
*TDH*	*Tdh*	*Tdh*	ENSOCUG00000011294
*TMED11P*	*Tmed11*	*Tmed11*	
*TMEM198B*	*Tmem198b*	*Tmem198b*	ENSOCUG00000027671
*TMEM30CP*	*Tmem30c*	*Tmem30c*	ENSOCUG00000027470
*UOX*	*Uox*	*Uox*	ENSOCUG00000027397
*ZNF271P*	*Zfp35*	*Zfp35*	ENSOCUG00000029705

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
