# Peer review of "Protein-Coding Genes in Euarchontoglires with Pseudogene Homologs in Humans"

_life, 2020, doi:10.3390/life10090192_

Round 1
Reviewer 1 Report
In the manuscript, the authors propose a new method for the detection of those pseudogenes that became pseudogenes relatively recently.
Generally, the correct prediction of pseudogene is an important problem. Unfortunately, the manuscript is not clear enough to assess the virtues of new algorithm.
First, the authors do not provide any context information. Specifically, the authors do not describe in any way other algorithms that exist for predicting such pseudogenes.
Second, they do not specify the advantages of the method they suggest. Their algorithm, in fact, is not described. The description of the algorithm is replaced by a short statement that the information about nearby genes is used. It appears that the suggested approach is based on the assumption of certain conservation of gene neighborhoods (although I cannot be sure). Meanwhile, as I understand it, it is the algorithm that is the main subject of the article.
The main results of the study are presented in the Supplementary File 1, which makes the reading very troublesome.
Therefore, the authors should re-write their paper, merge the Supplementary File 1 with the main text, provide some information on other, competing approaches to the problem. The new approach should be discussed in detail, so that the reader should be able to understand it and make an informed decision on whether to use the new method or not. The new approach should be discussed in relation to previous contributions to the subject. Generally, the authors should invest efforts into making their contribution clear and understandable.
Author Response
The authors are sincerely grateful to the reviewer for their valuable and useful comments and suggestions and have tried to fit all answers within the frame of a Short Communication.
Answers to Reviewers’ Comments
- The authors do not provide any context information. Specifically, the authors do not describe in any way other algorithms that exist for predicting such pseudogenes.
Response. The proposed method is not designed for predicting de novo pseudogenes in addition to the manifold already known. Instead, we aim to identify relatively recent pseudogenes. The text cites several relevant works [9, 8], which contain further references. The authors are unaware of other algorithms providing for a large-scale solution of this task.
- They do not specify the advantages of the method they suggest. Their algorithm, in fact, is not described. The description of the algorithm is replaced by a short statement that the information about nearby genes is used. It appears that the suggested approach is based on the assumption of certain conservation of gene neighborhoods (although I cannot be sure). Meanwhile, as I understand it, it is the algorithm that is the main subject of the article.
Response. The method is now more detailed in section 2 (Materials and Methods) and illustrated with Figure 2. Relevant parameters are empirically estimated and not detailed here. We assume that a gene’s survival relates to maintaining the synteny of its neighborhood at a certain conservative level.
- The main results of the study are presented in the Supplementary File 1, which makes the reading very troublesome.
Response. Results now include a reduced version (Table 1) of supplementary Table S1 with a detailing text. The manuscript is formatted as a short communication to reflect current stage of the research.
Reviewer 2 Report
In the manuscript "Protein Coding Genes in Euarchontoglires with Pseudogene Homologs in Human" by Rubanov et al, the authors report the development of a fast software for the detection of inactivated genes in different species.
While the title of the manuscript sounds exciting, I am disappointed to say that I found it very hard to follow the results of the manuscript. Firstly, the description of the methodology i.e. the Materials and Methods section is highly inadequate. It is absolutely not clear to me if the gene loss detection software that the authors have developed relies on existing gene annotations or it screens the sequences from genome alignments or builds alignments by using the sequence of the gene from the reference species. Secondly, the authors also do not specify the kind of pseudogenes they are interested to find -- processed/unprocessed/unitary.
Since the methodology lacks clarity, I found it very hard to follow the results.
On a related note, the authors should also try to change the style of the manuscript and try to make use of phrases and sentences that are not ambiguous. For instance: "We searched for protein-coding genes in reference species, mouse (Mus musculus), rat (Rattus norvegicus) and rabbit (Oryctolagus cuniculus)" (line 37), I do not understand what is the reference species here. Is it only mouse or is it mouse, rat and rabbit?
Similarly in the sentence (line 33) "The challenge is to detect pseudogenes in an arbitrary species (in our case, human) or a set of species based on ancestry patterns of protein-coding homologs in an arbitrary species set (here, the Euarchontoglires group)" the authors should make use of the word 'query' instead of 'arbitrary'.
I would also appreciate if the authors could devote a Figure that shows the phylogenetic tree and indicates the lineages that were queried to discover gene-loss events together with the reference species.
Lastly, the authors should try to clarify the methods and describe those in greater detail. This could be achieved by perhaps, drawing a schematic and submitting it as a Supplementary figure though this does not imply that the text in the Materials and Methods does not need improvement.
Author Response
The authors are sincerely grateful to the reviewer for their valuable and useful comments and suggestions and have tried to fit all answers within the frame of a Short Communication.
Firstly, the description of the methodology i.e. the Materials and Methods section is highly inadequate. It is absolutely not clear to me if the gene loss detection software that the authors have developed relies on existing gene annotations or it screens the sequences from genome alignments or builds alignments by using the sequence of the gene from the reference species.
Response. Materials and Methods are essentially restructured and extended. A detailed description of the software hardly fits a short communication and is partly provided in [11]. The method and software for pseudogene prediction utilize orthology and genomic alignments currently retrieved from Ensembl.
Secondly, the authors also do not specify the kind of pseudogenes they are interested to find -- processed/unprocessed/unitary.
Response. We equally treated all pseudogene types. All predicted pseudogenes are listed with their types in Table S1 (column F). Totals of pseudogenization events per type are provided in Results (lines 105--108). The method allows detection of recent pseudogenizations of any predefined type.
The authors should also try to change the style of the manuscript and try to make use of phrases and sentences that are not ambiguous. For instance: "We searched for protein-coding genes in reference species, mouse (Mus musculus), rat (Rattus norvegicus) and rabbit (Oryctolagus cuniculus)" (line 37), I do not understand what is the reference species here. Is it only mouse or is it mouse, rat and rabbit?
Response. Initially, searches were conducted separately for each reference species, and the inferred gene lists were united afterwards, as is now described in more detail in Materials and Methods.
Similarly in the sentence (line 33) "The challenge is to detect pseudogenes in an arbitrary species (in our case, human) or a set of species based on ancestry patterns of protein-coding homologs in an arbitrary species set (here, the Euarchontoglires group)" the authors should make use of the word 'query' instead of 'arbitrary'.
Response. Corrected.
I would also appreciate if the authors could devote a Figure that shows the phylogenetic tree and indicates the lineages that were queried to discover gene-loss events together with the reference species.
Response. Descriptive Fig. 1 added.
Lastly, the authors should try to clarify the methods and describe those in greater detail. This could be achieved by perhaps, drawing a schematic and submitting it as a Supplementary figure though this does not imply that the text in the Materials and Methods does not need improvement.
Response. Descriptive Fig. 2 added schematically explaining the method.
Reviewer 3 Report
The manuscript entitled “Protein Coding Genes in Euarchontoglires with Pseudogene Homologs in Human” deals with automated computer-aided search for human pseudogenes corresponding to protein-coding genes in apes and glires (Euarchontoglires). A large-scale search revealed only 50 such pseudogenes, part of theirs is largely correspond to the genes associated with human pathologies and became nonfunctional (see the references in the ms). This can be considered as an algorithm testing although the other part of human pseudogenes (as protein-coding in other apes and glires) were identified by the program for the first time. These are potential targets for in-depth biological and/or medical studies. The identified genes include Zfp35 associated with asthma, Fbxl21 associated with circadian rhythm disorders, Ahsa2, Tmem198b, etc. The RGD1560171 gene can be linked to longevity and so on.
Such a small number of human pseudogenes (50) is unexpected and intriguing. Another interesting finding is that some of the identified genes pseudogenized in humans and exactly one more ape species. This suggests similar consequences of the pseudogenization, which are indeed observed in their ontogeny.
All results in the manuscript are based on the per-gene co-verification of local synteny, similarity of exon-intronic structures, and orthology in Euarchontoglires. The developed method is applicable to large sets of complete genomes (over 100). It makes possible finding a list of pseudogenes in any single species (or at species set) that are protein-coding genes at other species set. This can be advantageous for bioinformatics studies on animals rather than humans. The program and user manual are available via the Internet [11]. The verifying the program is out of the question.
The authors provided no statistical testing of the program, which is desirable but probably not possible in a short communication. Such testing would raise challenging problems of generating an adequate sample and of the validity test. No program description is available in the text, which is also natural for short communication.
The manuscript agrees with the role of an evolutionary recent function change of a gene and its pseudogenization, in particular, for human physiological processes (soundly established in recent decades).
No reputable software for the search of pseudogenized protein-coding genes is currently available although identified human pseudogenes are numbered in thousands. Thus, this short communication presents one of the first such programs.
One minor criticism: “human-like” in line 79 seems inept and should be omitted.
Overall, the manuscript deserves publication as a short communication.
Author Response
The authors are sincerely grateful to the reviewer for their valuable and useful comments and suggestions and have tried to fit all answers within the frame of a Short Communication.
One minor criticism: “human-like” in line 79 seems inept and should be omitted.
Response. Corrected (line 117).
Reviewer 4 Report
In this paper Rubanov and colleagues show an original bioinformatics technique to identify protein coding genes in rodents, lagomorphs and nonhuman primates that pseudogenized in human. The paper is fine and, in my opinion, it needs only minor revisions. Thus I advice the authors to show some cartoons that can help also the non-specialized readership to better comprehend their proposed method.
Author Response
The authors are sincerely grateful to the reviewer for their valuable and useful comments and suggestions and have tried to fit all answers within the frame of a Short Communication.
I advice the authors to show some cartoons that can help also the non-specialized readership to better comprehend their proposed method.
Response. The method is now clarified in extended Materials and Methods, including Figs. 1--2, supplemented text and Table 1 with narration.
Round 2
Reviewer 1 Report
The manuscript can be published in the present form